# The Agreement between Radiography and Fluoroscopy as Diagnostic Tools for Tracheal Collapse in Dogs

**DOI:** 10.3390/ani13091434

**Published:** 2023-04-22

**Authors:** Wasutorn Yangwanitset, Somkiat Huaijantug, Mookmanee Tansakul, Walasinee Sakcamduang

**Affiliations:** 1Faculty of Veterinary Science, Mahidol University, Nakhon Pathom 73170, Thailand; wasutorn.yan@gmail.com; 2Department of Clinical Sciences and Public Health, Faculty of Veterinary Science, Mahidol University, Nakhon Pathom 73170, Thailand; somkiat.hua@mahidol.ac.th (S.H.); mookmanee.tan@mahidol.ac.th (M.T.)

**Keywords:** diagnostic imaging, dog, fluoroscopy, radiography, tracheal collapse

## Abstract

**Simple Summary:**

Tracheal collapse is a common disease in small breed dogs, and imaging techniques such as radiography and fluoroscopy are important for diagnosis. However, there is limited agreement between these two methods in different regions of the trachea. A study was conducted on 29 dogs with tracheal collapse to investigate the agreement between thoracic radiography and fluoroscopy. The results showed that radiography underestimated the degree of collapsing trachea compared to fluoroscopy significantly at carina area. However, the agreement between the methods was found to be only slight at the cervical region. The study suggests that while radiography can be useful for screening, fluoroscopy can detect a greater degree of collapsing trachea than the two-image radiography method.

**Abstract:**

Tracheal collapse is a common disease in small, middle-aged dogs, and imaging tools are essential for its diagnosis. Radiography and fluoroscopy are the main diagnostic modalities used, but their agreement in different regions is not well documented. In this study, the agreement between thoracic radiography and fluoroscopy in tracheal collapse was investigated in 29 dogs. The results showed that radiography detected a lower degree of collapsing trachea compared to fluoroscopy at the carina region (*p* < 0.001). However, there was no significant difference observed between the degree of collapsing trachea detected by radiography and fluoroscopy at the cervical, thoracic inlet, and intra-thoracic regions (*p* = 0.780, 0.537, and 0.213, respectively). The kappa statistic indicated a slight agreement at the cervical region at a 16.4% cut-off (κ = 0.20), while the other regions showed a non-agreement. In conclusion, although radiography is useful for screening, fluoroscopy was able to detect the degree of the collapsing trachea greater than radiography in the carina region. Additionally, if a collapse in the cervical region is detected by radiography, it is prone to have a positive relationship with fluoroscopy as well.

## 1. Introduction

Tracheal collapse is a typical respiratory disease in middle-aged, small dogs. The cause of the disease is still unknown [1]. The clinical manifestations of tracheal collapse vary in accordance with the disease’s severity, ranging from being asymptomatic to the occurrence of a paroxysmal dry cough that sounds like a “goose honk,” exercise intolerance, cyanosis, and respiratory distress [1].

The dynamic changes in the tracheal lumen during the inspiratory and expiratory phases indicate the severity of the disease [1]. During the inspiratory phase, the extrathoracic segment of the trachea tends to collapse due to more negative intraluminal pressures. Conversely, during the expiratory phase, the intrathoracic segment is more likely to collapse because of the increased intrapleural pressure [2]. Therefore, many imaging techniques, such as radiography, fluoroscopy, computed tomography (CT), ultrasound, and bronchoscopy, were useful to diagnose the disease [3,4,5,6,7].

Thoracic radiography is a common tool that is used to confirm this disease. However, it shows inconsistent results that have a sensitivity of around 45–86%, depending on the tracheal regions [3,6]. Fluoroscopy is a useful tool for investigation because it can detect the dynamic change in both respiratory phases [8]. However, one study showed that fluoroscopy had a lower sensitivity than radiography, which is potentially due to its tendency to produce low-resolution images in order to minimize radiation exposure [4]. Nevertheless, the level of agreement between radiography and fluoroscopy was moderate to good during tidal respiration and cough [4].

In human medicine, the gold standard for detecting central airway collapse is endoscopy [8,9,10]. Additionally, in veterinary medicine, it allows veterinarians to observe within the lumen and collect samples for further investigation, such as bacterial culture. However, this technique must be performed under general anesthesia, which might not be suitable for some dogs, especially those with a high ASA score [11].

Currently, there is yet to be a consensus on a cut-off point for classifying the degree of this disease [12]. However, the most common scheme for grading this disease was defined by Tangner and Hobson in 1982, using a percent reduction in the lumen separated into four grades obtained by bronchoscopy: grade I is defined as a percentage change less than approximately 25%, grade II is 25–50 percent, grade III is 50–75 percent, and grade IV is 75–90 percent [13]. Additionally, the study in 19 small (<10 kg) healthy dogs using the right lateral recumbency showed that the variation of tracheal height was around 0–16.4% and 0–13.5% in radiography and fluoroscopy, respectively [14].

Although there was a good level of agreement between fluoroscopy and radiography when considering the entire trachea, M. Macready’s research has revealed significant differences in the percentage change in the thoracic inlet, intrathoracic, and carina regions [4,7]. This suggests that the agreement may vary depending on the specific regions examined, emphasizing the importance of studying each region separately. Currently, there is limited research on the agreement between radiography and fluoroscopy in each specific region. Our hypothesis was that there would be an agreement between these two modalities only in the cervical region. The objectives of this study were to determine the agreement between these two diagnostic methods in four distinct regions of the trachea and observe the response to treatment of dogs throughout the study.

## 2. Materials and Methods

### 2.1. Dogs 

Dogs were included in this prospective study using the following criteria: (1) those that showed a history of or presented related clinical signs of tracheal collapse, such as cough, gagging, exercise intolerance, and respiratory distress; (2) those with a confirmed disease by imaging study from other hospitals. Dogs were excluded from the study if they did not have a collapsing trachea detected obtained by fluoroscopy (tracheal diameter change < 16.4% in any region), as evaluated by WY, who has 5 years of experience in small animal clinical practice, and confirmed by SH, who has 25 years of experience in small animal diagnostic imaging. Additionally, dogs were excluded if they had non-compliance or loss of follow-up. Both the imaging study and clinical evaluation were conducted at Prasu Arthorn Veterinary Teaching Hospital, Faculty of Veterinary Science, Mahidol University. Informed consent was obtained from the owner before enrolment. The animal ethics for this study were approved by the Faculty of Veterinary Science, Mahidol University-Institute Animal Care and Use Committee No. MUVS-2020-01-02. The clinical signs of the dogs were monitored on days 14, 28, and 56 using a cough symptom evaluation questionnaire modified from Jeung et al. (Appendix A) [15].

### 2.2. Imaging Study 

Inspiration and expiration thoracic radiographs in right lateral recumbency were obtained from non-sedated dogs using a digital radiographic system (Bucky Diagnost CS, Philips Medical Systems, Hamburg, Germany). When patient compliance was possible, fluoroscopy was performed with dog positioned in right lateral recumbency during tidal respiration at least 3 respiratory cycles. Images at full inspiratory and expiratory phases obtained from a fluoroscopy machine (Philips BV Libra, Hamburg, Germany) were captured. Percent changes in trachea greater than 16.4 were classified into presence of tracheal collapse, while values lower than these were classified as normal fluctuation of trachea [14]. The location of trachea was divided into 4 areas, including cervical (C4–C6), thoracic inlet (C7–T1), intra-thoracic (T2–T4), and carina (Figure 1). The percentage difference in tracheal height between full inspiration and full expiration was measured by ImageJ software and calculated by formula: % tracheal height = ((mean tracheal height at maximum inspiration − mean tracheal height at max expiration)/mean tracheal height at maximum inspiration) × 100 [13,14]. Both radiographic and fluoroscopic studies were performed at 3 different times (days 0, 14, and 56). Three sets of imaging were measured by WY under the supervision of SH, MT, and WS and were reported together in overall analysis (Figure 2).

### 2.3. Statistical Analysis

Computerized statistical software (SPSS 18.0 for Windows, Chicago, IL, USA) was used for analyses. The numerical data were tested normal distribution using Shapiro–Wilk test. The diameter change in each region between 2 methods was tested by paired t-test or Wilcoxon signed rank test, as appropriate. The agreement between 2 methods was calculated by McNemar’s test or binomial distribution as appropriate. The level of agreement was analyzed and interpreted by Cohen’s kappa coefficient [16]. *p* < 0.05 was considered statistically significant.

## 3. Results

### 3.1. Demographic Characteristics and Clinical Outcomes

Fifty-three tracheal collapse dogs were enrolled in this study. Twelve dogs were excluded from the study due to no detection of dynamic change in the trachea in the first visit (*n* = 9), non-compliance (*n* = 1), the loss of follow-up (*n* = 1), and detected diarrhea during the study due to dietary changes (*n* = 1). Unclear boundary images of the trachea and/or partial invagination or redundant dorsal tracheal membrane image detection were excluded before analysis (*n* = 12). Three sets of twenty-nine dogs were analyzed. There were 15 male and 14 female dogs (6 neutered males and 11 spayed female dogs). The breeds included 12 Pomeranians (41.4%), 7 Pugs (24.1%), 4 Chihuahuas (13.8%), 3 Shi-Tzu (10.3%), 2 Yorkshire Terrier (6.9%), and 1 English Mastiff (3.4%). The Mean ± SD of age was 7.6 ± 3.7 years old, and the median (interquartile range, IQR) weight was 6.1 (3.7–8.6) kg. The median of BCSs (9-grade system) was 7/9 (IQR 5–8). Treatment at each visit was reported in Appendix A. Results of the cough symptom evaluation questionnaire showed that all dogs had improvement in the interval variable, but there was no significant difference in the duration of cough (Appendix A).

### 3.2. Imaging Results

The findings showed no significant difference in the percent change in tracheal diameter between radiography and fluoroscopy in all regions except for the carina region. Throughout all the time periods, the fluoroscopy was able to detect a significantly greater degree of collapse compared to radiography (overall analysis; *p* = 0.001, day 0; *p* = 0.037, day14; *p* = 0.010 and day 56; *p* = 0.002; Table 1).

The results of McNemar’s test showed that when using a 16.4% cut-off, there was no statistically significant difference between the results obtained from the fluoroscopy and radiography at the cervical, thoracic inlet, and intrathoracic regions (*p* = 1.000, 0.522, and 0.184, respectively; Table 2). However, the level of agreement between the two methods, as measured by the kappa statistic, was low, with only the cervical region showing a slight agreement (κ = 0.20; Appendix A). Additionally, the level of agreement varied by time point, with the highest agreement observed on day 0 and the lowest agreement observed on day 14 (κ = 0.47 and 0.17, respectively; Appendix A).

## 4. Discussion

This study suggested that fluoroscopy can detect diameter changes significantly differently from radiography, particularly in the carina region. Notably, the the findings of the study were in agreement with the results of a previous study at the carina and the cervical regions, but not at the thoracic inlet and the intrathoracic regions [7].

This result from the agreement test contrasts with our hypothesis that the two techniques should have moderate agreement. Cohen’s kappa analysis, using fluoroscopy as a reference tool, revealed a slight degree of agreement between the two techniques in the cervical region, while the other regions showed no agreement between the two tests. Interestingly, on day 0, although the binomial distribution showed a non-concordant result between both tests, the kappa statistics showed a moderate agreement (κ = 0.47; see Appendix A). This might have resulted from the percent agreement on day 0, which was the highest (79.31%) compared to the other visits (72.41%; both on days 14 and 56). It can be concluded that the level of agreement was influenced by the size of the airway lumen. Therefore, in cases of suspected airway collapse, particularly in the region spanning from the thoracic inlet to the carina, the use of fluoroscopy or bronchoscopy should be contingent upon the owners’ consent and a careful assessment of the potential risks and benefits associated with the procedure.

One limitation of this study is that the statistical calculations made on the same subjects in different periods may not be representative of the population. However, the power of the test depends on the sample size, which influences its ability [17]. Therefore, this study decided to find the agreement over time. Nevertheless, it could still be useful for imaging follow-up purposes. Furthermore, only two images of radiography were performed per visit, without replications as can be conducted with fluoroscopy, which could have affected the measurement of tracheal diameter during both respiratory phases. Remarkably, this study did not perform bronchoscopy as the gold standard to investigate this disease. Hence, the diagnostic accuracy might be less reliable compared to bronchoscopy. However, the effect of anesthesia on tracheal collapse should also be considered, as it can obscure contributing factors such as excitement.

## 5. Conclusions

The screening of suspected tracheal collapse in dogs can be effectively performed using radiography. However, when compared to radiography, fluoroscopy can detect a greater degree of tracheal collapse at the carina region. It is important to note that if radiography shows collapse in the cervical region, which is the largest segment of the trachea detected during inspiration, it is more likely that the dog has extra-thoracic collapse, which is similar to fluoroscopy. For the thoracic inlet, intra-thoracic, and carina regions, it is recommended to further investigate using fluoroscopy, CT, or bronchoscopy.

## Figures and Tables

**Figure 1 animals-13-01434-f001:**
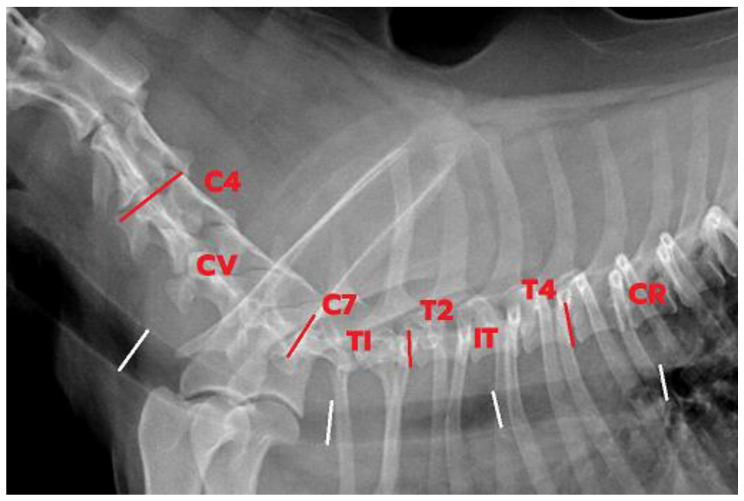
Four measurement locations (red lines) of tracheal height at inspiration phrase vertebra as a reference landmark and measurement size (white lines): cervical (CV; C4–C6), thoracic inlet (TI; C7–T1), intra-thoracic (IT; T2–T4), and carina (CR).

**Figure 2 animals-13-01434-f002:**
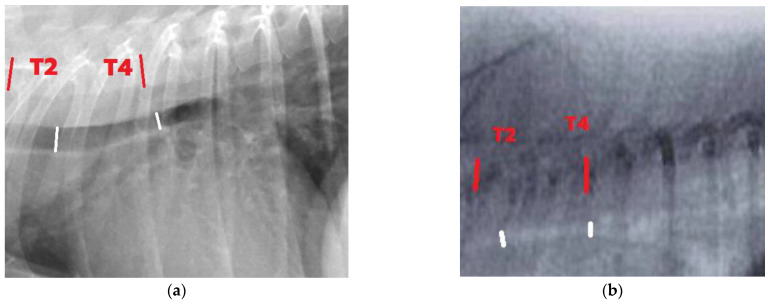
The following images (**a**,**b**) were acquired from a single dog during the same visit, with (**a**) being a radiography image and (**b**) being a fluoroscopy image captured during the expiration phase. The red line was located in the intra-thoracic region between T2 and T4, while the white line was situated at the measurement point of the trachea in the intra-thoracic and carina region.

**Table 1 animals-13-01434-t001:** Median and interquartile diameter change percentages were reported separated by fluoroscopy and radiography. Wilcoxon signed rank test showed significant difference in carina region, while the 3 other regions showed no significant differences.

	Region	Fluoroscopy (%)	Radiography (%)	*p*-Value
Overall	Cervical	8.04 (4.32–14.02)	9.29 (3.29–15.04)	0.780
	Thoracic	12.35 (8.43–17.97)	11.33 (4.2–20.4)	0.537
	Intra thoracic	12.46 (8.86–20.13)	12.20 (6.73–17.6)	0.213
	Carina	18.75 (14.12–27.31)	11.59 (4.76–20.15)	<0.001
Day 0	Cervical	11.11 (5.40–21.28)	10.06 (3.8–13.06)	0.243
	Thoracic	15.52 (10.00–20.20)	14.79 (7.02–21.83)	0.905
	Intra thoracic	15.00 (10.94–21.74)	12.84 (6.43–17.44)	0.090
	Carina	20.38 (14.31–27.92)	13.86 (6.51–22.28)	0.037
Day 14	Cervical	8.04 (4.12–13.09)	6.54 (2.08–15.99)	0.974
	Thoracic	9.97 (7.37–17.21)	10 (3.925–18.42)	0.627
	Intra thoracic	13.24 (6.95–19.62)	15.32 (10.43–20.35)	0.347
	Carina	20.10 (13.74–27.90)	11.22 (4.44–17.15)	0.010
Day 56	Cervical	7.17 (3.65–12.09)	10.39 (3.66–15.37)	0.150
	Thoracic	10.39 (8.54–16.60)	9.93 (2.88–21.00)	0.611
	Intra thoracic	11.89 (7.97–19.76)	8.49 (5.04–13.36)	0.150
	Carina	17.01 (13.47–24.16)	11.46 (4.08–18.85)	0.002

**Table 2 animals-13-01434-t002:** Results of agreement test between fluoroscopy and radiography were reported with kappa statistic.

Regions	Agreement (%)	Both Present	Both Absent	Flu+/Rad−	Flu−/Rad+	*p* Value of McNemar’s Test	Kappa	Odds Ratio (CI)
Cervical	65/87 (74.72)	6/87	59/87	11/87	11/87	1.000 *	0.20	1.0 (0.39–2.54)
Thoracic inlet	48/87 (55.17)	10/87	38/87	17/87	22/87	0.522	0.00	1.3 (0.66–2.60)
Intra thoracic	41/87 (47.13)	7/87	34/87	28/87	18/87	0.184	−0.15	0.6 (0.34–1.20)
Carina	40/87 (45.98)	19/87	21/87	35/87	12/87	0.001	−0.01	0.3 (0.16–0.68)

* Binomial distribution was calculated due to discordant pairs is less than 25

## Data Availability

The data used to support the findings of this study are available from the corresponding author upon request.

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
