# Peer review of "The Agreement between Radiography and Fluoroscopy as Diagnostic Tools for Tracheal Collapse in Dogs"

_animals, 2023, doi:10.3390/ani13091434_

Round 1

Reviewer 1 Report

Dear Authors,

it has been my pleasure to read your communication titled “ Tracheal collapse diagnostic tools agreement in dogs”.

Despite the topic of the research, which is potentially interesting, I found your paper confusing and partially lacking of novelty. As you have cited in your references, papers aiming to compare radiographs and fluoroscopy for the detection of tracheal collapse in dogs have already been published, and with higher number of cases.

Apart from this, I found some major flaws in the study design: if you want to compare two methods (xrays and fluoro) and decide which one performs best, you should have a gold standard (bronchoscopy?). If you use fluoroscopy as a gold standard you can only test radiograph’s performances but you cannot state that fluoroscopy is superior.

Second, the results are reported in a very confused way, it is not clear why you assessed the same cases three times (then it seems that all numbers go together for the statistical analysis?). It would be an idea to focus the attention only on the follow up. What is this treatment that you used that reduced the degree of collapse? Was it evident both at radiographic and fluoroscopic examination? How better was the degree of collapse and how this relates with the clinical signs?

Please find some more precise comments below.

Simple summary and abstract: in both summaries it is said that there is “consistency” of the two methods. What do you mean by consistency? I found this in contradiction with the non-agreement stated one line after.

Introduction: it is not clear from your introduction what is the background and what are the aims of your study. I would consider re-write it explaining more the objectives of the study. What was your hypothesis? What did you want to test?

Line 32: I am not sure tracheal collapse and tracheomalacia can be used interchangeably.

Line 35 and 39: I would consider changing the term beneficial. Useful is maybe more appropriate?

Line 42: please rephrase (should be maybe in veterinary medicine?)

Line 49: percentage of reduction?

Line 51: the study

Materials and Methods: I would include a paragraph with the study design as it is not clear. Was it a retrospective study? Prospective? Did you enroll patients based on the clinical suspicion or imaging results? Inclusion criteria must be listed

Line 66: I would use recumbency instead of position

Line 81: you performed three times the same and then put all the results together? Please clarify

Line 88: I would use one method for the agreement.

Line 90: you tested the accuracy etc of radiographs only, using fluoroscopy as gold standard

Line 94: p < 0.05

Results: this section is really confusing. I would try to simplify the results (starting with a simpler study design and objectives). You can use tables but all together they are not explaining the key points of your work.

Line 114-118: overall analysis showed concordance but then no agreement?

Line 120-125: as already mentioned, I think you need a third test to evaluate the accuracy. Or, if you already take fluoroscopy as gold standard, you cannot conclude that fluoroscopy is superior to radiography.

Discussion

Line 136-137: see other comments, your results are not saying that fluoroscopy is superior

Line 152-154: I thought that you excluded these patients (cfr lines 100-102), so how can they influence the sensitivity?

Line 154: I think this is an interesting point, in my experience the medical treatment is not that effective while in your cases it seems that it changed the imaging findings. Could you explain which treatment? How much the collapse resolved? With both modalities?

Line 169: I think there are more limitations

Line 173: Again, you tested radiography based on the fact that fluoroscopy is superior, this cannot be the conclusion of your study.

Figures: I would like to have some radiographic and fluoroscopic images with comparison of the same dog to better understand your findings.  

Author Response

Dear Reviewer 1,

We have addressed the main recommendations and included a point-to-point response. We have also made changes to the manuscript as suggested and highlighted them in yellow. However, we have uploaded the manuscript with highlighted yellow, blue, and green, based on the respective recommendations of reviewers 1,2, and both/Journal Guidelines respectively.

Best Regards,

Walasinee

Reviewer 2 Report

This study evaluated the agreement between radiography and fluoroscopy. Authors provided adequate study information in content. However, the study method and discussion could be strengthened better after adding required information.

Title: Since only 2 diagnostic tools were used, it is recommended to provide the name of the tools in title.

Line 10: please revise “be-tween” to “between”.

Line 35: please list the name of the “many imaging techniques”.

Line 54-56: please declare the advantages of this study when compared to previous studies.

Line 57: prospective or retrospective? Blind or unblind study? Please provide the the background of the image evaluators.

Line 59: Were the clinical signs noted?

Line 67-70: Information of imaging modalities, such as brand, company or location,  should be provide.

Line 70-72: The level of tracheal collapse was list. However, you did not apply the criteria to analyze the results. Consider leaving the necessary information only in this section or remove to introduction.

Line 76: loss of dot behind trachea.

Line 95: This study apparently is classified as an imaging study. As a result, it is highly recommended to provide reference images in the content.

Line 97-102: The exclusion criteria should be added in section of materials and methods. It is weird that you excluded the dogs without dynamic change. They were still dogs diagnosed by trachea collapse according to line 59. Alternately, you can explain the reason why they were excluded here.

Line 107: (5,8) means range or interquartile?

Line 119: According to your aim of this study, comparing the imaging tools was the main thought. Therefore, the two image tools should be put on an equal level. Is fluoroscopy recognized as a golden diagnostic tool for canine trachea collapse? If not, the sensitivity and specificity is not adequate for using as study methods. If so, please add the information in introduction and revise the study structure.

Line 156-162: The discussion of sensitivity and specificity should also be reconstructed according to the mentioned above.

Line 168: Please explain why you acquired images for 3 periods here or in methods. Do the dogs receive any treatment during the period?

Line 168: Could you add more description about why the fluoroscopy did not “overestimate” the results?

Line 168: Please compare the different imaging modalities for trachea collapse, such as CT, bronchoscopy…

Line 168: Consider add the advantages and disadvantages of the two modalities.

Line 169: I believe there were still other factors may interfere the accuracy, such as only one imaging view was taken, exclude the dogs without tracheal dynamic changes.

Line 173-174: Please add valuable information for clinician, such as how to select tools in practice, or the indication when they interpret the images from different modality.

Author Response

Dear Reviewer 2,

I will submit a revised version of the manuscript upon suggestions from you and the other reviewer to the editor.

Best Regards,

Walasinee

Round 2

Reviewer 1 Report

Dear Authors, 

thank you for your corrections to the manuscript " The agreement between radiography and fluoroscopy as diagnostic tools for tracheal collapse in dogs". I believe the quality of the paper overall improved from last version. 

I still have some comments: 

Introduction: line 63-64 you have to specify diagnostic ability of radiography. It would be worthy to include also the response to treatment as aim of the study, for me this remains the point that is more interesting (and new) of your article. Otherwise, it is not clear why you repeated the imaging at different time points. 

Materials and methods: the inclusion criteria are still missing, only one exclusion is described. I assume the inclusion were clinical signs and fluoroscopy consistent with tracheal collapse, but I would suggest to specify. 

Line 91: I would put the citation of Figure 1 here. 

Results: the paragraph about imaging results is still confusing and I find many of the tables included not necessary. I would suggest to remove the supplementary material if not specified otherwise by the Editor. I would suggest to summarise your major findings without using all these tables. 

Citation of Figure 2? 

Discussion: I always have the same problem with your wording. The fact that fluoroscopy can detect more changes is (maybe) because you assume fluoroscopy is always right. Is it possible that some of your fluoro were false positive? That you overestimated the degree of collapse? This has to be taken into consideration before concluding, as you don't have a third method (bronchoscopy). 

Lines 215-216: Two radiographs, I don't think we can call them frames as it is not a dynamic or continuous acquisition.

Conclusion: once again, you cannot say fluoroscopy is more efficient based on your results. 

Line 229: superior techniques based on previous studies? I would suggest to say cross-sectional or, in case you refer to endoscopy, to say which techniques it is advised to use. 

Lines 37, 207: I think computed tomography is the correct word for CT. If you spell it once then you can use CT

Figures: 

In Figure 1 you don't show the measurements, just the locations. So it is useless to include both inspiration and expiration, as the vertebrae will be likely the same. 

In Figure 2 you have to specify that is the same dog at the same time point and show where or how the measurements were different between the two modalities. 

Supplementary material: I find discrepancy between what is listed and what is provided. Nevertheless, I think the only tables you should include are the ones with the measurements and the ones with the agreement. I would not include ROC curves and sensitivity/specificity of radiography as you used fluoroscopy as gold standard when it is not always the case. I would limit your observations on the agreement between the two methods, without concluding on which is best because you cannot tell.  (points 2 and 3 of your objectives are for me not completely assessable/correct with your study design). 

Author Response

Dear Reviewer 1,

We are pleased to submit the revised version of our manuscript entitled "Tracheal Collapse Diagnostic Tools Agreement in Dogs" for publication consideration in Animals. We appreciate the time and effort you and the reviewers have put into evaluating our manuscript, and we have carefully considered all the comments and suggestions provided.

We have addressed the main recommendations of the reviewers, and have included a point-by-point response to their comments in the revised manuscript. We have also made changes to the manuscript as suggested, and highlighted them in yellow, blue, and green, based on the respective recommendations of reviewers 1, 2, and both/Journal Guidelines respectively.

Sincerely,

Walasinee

Reviewer 2 Report

Thank you for the revising. However, the important issues of this study still remained. First, the reason or necessary of multiple imaging examinations was not clear. Second, the using of fluoroscopy as golden standard to obtain the results of sensitivity and specificity was considered inappropriate in this study. Since the article is classified as communication type, it is recommended to remove these contents.

Line 42-45:

Add a reference behind “…a lower sensitivity than radiography.”.

Provide the brief possible reason of the phenomenon to improve the article flow. I believe you can find it from the reference [3].

Line 59-64:

The flow of this paragraph needs to be rewrite extensively. Simple is better than multiple but nonlogical.

Although you combined two research methods for this study, you still have to provide the adequate information of its benefits or disadvantages of previous studies.

I am confused about your hypothesis. It is difficult to understand about its basis. Why it should be consistent? What will be consistent?

“…four distinct regions of the trachea…” The information of “four distinct regions” should be added before here.

Line 67-74

The inclusion criteria and exclusion criteria were confused. Since dogs with trachea collapse could be recruited only via radiography, it was believed dogs were excluded after formal imaging study rather than “before”. It is better to add necessary information here.

Line 82, 86

The information of facilities in article is commonly written as (example) “…using a digital radiographic system (Bucky Diagnost CS, Philips Medical Systems, Hamburg, Germany) ”

Line 87-90

The grading I was confused. Why don’t you use 10-25 percent as grade I?

Why you list grade system in methods? Was the grade system evaluated via radiography or fluoroscopy? More, the associated results still could not be found in study. All methods should have their own results in scientific article otherwise they were believed unnecessary or results missed.

Line 95-97, 150-155, 183-202

Since the study experienced a period rather than once data collecting, the any intervention during the study (ex. medical intervention or do nothing) should be addressed here.

Besides, the reasons of multiple images acquiring still can not be understanded and they should be added or discussed if you insist to keep them in this study.

Line 105-106

The sensitivity and specificity were still considered incorrect here although you mentioned that the method was applied by Macready (2007). Macready recruited the dogs with trachea collapse which were diagnosed by fluoroscopy "with" cough, then compared with radiography. Therefore, fluoroscopy was reasonable recognized as golden standard in that study. Your experimental design was recruiting the dogs with possible trachea collapse, then evaluated by radiography and fluoroscopy "without" cough, then compared them. For these reasons, the fluoroscopy could not be recognized as golden standard in you study and two methods should be evaluated objectively. If you still would like to use the same method, the study design and flow should be rewritten.

Line 142-144

The lines in figure were positioned in vertebrae which were apparently not the location of measurement. It is recommended to put the key vertebrae number (C4, C7, T2, T4) on specific vertebrae bone and reference lines on trachea region. Any abbreviation used in figures (CV, TI, IT, CR) should be introduced in legend.

Line 146-147

Please mark the location of trachea.

Line 177

Why day 0 was different from others? Only “interesting” was not enough in discussion.

Line 180-182

“…based on individual patient risk”? The imaging modality was applied to diagnose the disease. How did you know the individual patient risk before the confirmation?

Line 229

What are small regions and superior techniques? The rough statements should be avoided.

Author Response

Dear Reviewer 2,

We are pleased to submit the revised version of our manuscript entitled "Tracheal Collapse Diagnostic Tools Agreement in Dogs" for publication consideration in Animals. We appreciate the time and effort you and the reviewers have put into evaluating our manuscript, and we have carefully considered all the comments and suggestions provided.

We have addressed the main recommendations of the reviewers, and have included a point-by-point response to their comments in the revised manuscript. We have also made changes to the manuscript as suggested, and highlighted them in yellow, blue, and green, based on the respective recommendations of reviewers 1, 2, and both/Journal Guidelines respectively.

Sincerely,

Walasinee
